# Impact of Postoperative Naples Prognostic Score to Predict Survival in Patients with Stage II–III Colorectal Cancer

**DOI:** 10.3390/cancers15205098

**Published:** 2023-10-22

**Authors:** Su Hyeong Park, Hye Seung Woo, In Kyung Hong, Eun Jung Park

**Affiliations:** 1Department of Hepatobiliary and Pancreatic Surgery, Severance Hospital, Yonsei University College of Medicine, Seoul 03722, Republic of Korea; kzpgg@yuhs.ac; 2Division of Colon and Rectal Surgery, Department of Surgery, Gangnam Severance Hospital, Yonsei University College of Medicine, Seoul 06229, Republic of Korea; hs6702@yuhs.ac (H.S.W.); on579@hanmail.net (I.K.H.)

**Keywords:** colorectal cancer, inflammation, surgery, Naples prognostic score, prognosis, survival

## Abstract

**Simple Summary:**

Systemic inflammatory markers are useful methods to predict the prognosis of colorectal cancer patients after surgeries. The Naples prognostic score is a useful predictive scoring system that reflects a patient’s inflammatory and nutritional status. In this study, the postoperative Naples prognostic score showed high accuracy in predicting survival after curative resection in stage II–III colorectal cancer patients compared with preoperative inflammatory markers.

**Abstract:**

Background: The Naples prognostic score (NPS) is a scoring system that reflects a patient’s systemic inflammatory and nutritional status. This study aimed to evaluate whether postoperative NPS is effective in assessing the prognosis of stage II–III colorectal cancer (CRC) patients compared with preoperative NPS. Methods: Between 2005 and 2012, a total of 164 patients diagnosed with stage II–III CRC, who underwent curative resection followed by adjuvant chemotherapy, were divided into two groups: Group 0–1 (NPS = 0–2) and Group 2 (NPS = 3 or 4). Preoperative NPS was calculated based on the results before surgeries, and postoperative NPS was assessed using the results obtained before adjuvant chemotherapy. Results: The overall survival of Group 0–1 was higher than that of Group 2 in both pre- and postoperative NPS assessments. According to the ROC curve analysis, the Area Under the Curve (AUC) ratio for postoperative NPS was 0.64, compared with 0.57 for preoperative NPS, 0.52 for the preoperative neutrophil–lymphocyte ratio (*p* = 0.032), and 0.51 for the preoperative platelet–lymphocyte ratio (*p* = 0.027). Conclusions: Postoperative NPS is effective in predicting the prognosis of stage II–III CRC patients who underwent curative resection followed by adjuvant chemotherapy. The use of NPS could be beneficial in evaluating the prognosis of CRC patients after surgeries.

## 1. Introduction

Systemic inflammation is known to impact tumor proliferation, induce angiogenesis, and contribute to the metastasis of primary cancer [1,2,3]. In the treatment of colorectal cancer (CRC), it is recognized that systemic inflammatory markers are useful for predicting a patient’s prognosis and survival after surgery [4,5]. Due to the role of inflammation in cancer patients, various inflammation-related biomarkers have been utilized to predict patient prognosis [6,7,8,9]. It is known that changes in carcinoembryonic antigen (CEA) levels have been used to assess cancer progression and recurrence [10,11]. However, neutrophil–lymphocyte ratio (NLR) [12], platelet–lymphocyte ratio (PLR), and lymphocyte–monocyte ratio (LMR) using systemic inflammatory markers before surgery have demonstrated usefulness in predicting clinical and oncologic outcomes after CRC surgeries. Additionally, since the patient’s immunity and nutritional status, such as prognostic nutritional index (PNI) and the controlling nutritional status (CONUT) score, influence the oncologic outcomes of CRC patients, a Naples prognostic score (NPS) including these factors was proposed to predict the prognosis of the patients [13,14].

The NPS is a scoring system that includes both systemic inflammatory and nutritional markers, such as serum albumin, total cholesterol, LMR, and NLR. It is considered to predict patient prognosis and long-term survival outcomes more effectively than previous systemic inflammatory markers. Galizia et al. [14] examined the relationship between preoperative NPS and postoperative patient prognosis in CRC. NPS was appropriate for predicting the survival rate of patients after surgery in CRC with high accuracy compared to previous systemic inflammatory markers. Miyamoto et al. [13] also analyzed that NPS was a useful tool for assessing the prognosis of metastatic CRC.

Preoperative systemic inflammatory markers and nutritional status of the patients were considered as the factors influencing the patient’s survival. However, postoperative outcomes following CRC surgeries are also crucial factors for survival and prognosis. Postoperative complications, pathologic characteristics of primary cancer, and adjuvant chemotherapy should be taken into account for patient survival and prognosis [15,16]. Therefore, it is necessary to consider postoperative recovery in addition to preoperative status when predicting the prognosis in CRC patients. This study aimed to evaluate the accuracy of postoperative NPS after CRC surgeries compared with preoperative systemic inflammatory markers and assess its effectiveness in predicting survival and prognosis in stage II–III CRC patients who are going to receive standard chemotherapy.

## 2. Materials and Methods

### 2.1. Study Population and Data Collection

This study aimed to retrospectively evaluate CRC patients who underwent surgery and adjuvant chemotherapy at Gangnam Severance Hospital, Seoul, South Korea, from January 2005 to December 2012. The study received approval from the institutional review board of our institution (IRB No. 3-2020-0484). A total of 299 patients diagnosed with stage II or III CRC and who underwent curative resection were included. Patients who did not receive adjuvant chemotherapy after curative resection or did not have blood test results after surgery before adjuvant chemotherapy were excluded. Finally, 164 patients were assessed in this study. According to the NPS, patients were divided into Groups 0–2 (Figure 1).

### 2.2. Definition of NPS

The NPS is a scoring system used to predict the prognosis of cancer patients [13,14]. NPS is calculated by four factors: NLR, LMR, serum albumin, and total cholesterol concentration. The neutrophil count, lymphocyte count, monocyte count, serum albumin concentration, and total cholesterol concentration were collected. A serum albumin level of ≥4 mg/dL was assigned 0 points, and <4 mg/dL was assigned 1 point. For total cholesterol concentration, patients with >180 mg/dL received 0 points, and those with ≤180 mg/dL received 1 point. Patients with an NLR of ≤2.96 were assigned 0 points, while those with >2.96 were assigned 1 point. An LMR > 4.44 was assigned 0 points, and ≤4.44 was assigned 1 point.

Patients were divided into three groups based on NPS scores: Group 0, Group 1, and Group 2. Patients in Group 0 had a total NPS score of 0. Group 1 had 1–2 points, and Group 2 had 3–4 points from the total NPS scores. Preoperative NPS was defined as the score calculated before surgery. On the other hand, postoperative NPS was defined as the score calculated before adjuvant chemotherapy after the recovery from CRC surgeries.

### 2.3. Evaluation Parameters

NPS was assessed using the results of blood tests collected before surgery and at the beginning of the first adjuvant chemotherapy. Regarding NPS and its role in predicting patient prognosis, this study included patients who had received adjuvant chemotherapy. Intraoperative outcomes, hospital stay duration, and the days until the commencement of adjuvant chemotherapy were evaluated to compare the differences between pre-operative NPS and postoperative NPS according to the groups. Postoperative complications were categorized based on the Clavien–Dindo classification [17], and tumor stages were assessed using tumor–node–metastasis (TNM) staging using the American Joint Committee on Cancer (AJCC), the 8th edition [18] for CRC.

### 2.4. Surgical Procedures and Adjuvant Chemotherapy

Patients with CRC stage II–III who underwent adjuvant chemotherapy after primary tumor resection were assessed in this study. They underwent open resection, laparoscopic operation, or robot-assisted operation, ensuring R0 resection. The choice of surgical methods was based on the surgeon’s preference and the location of the primary tumor. Adjuvant chemotherapy commenced within two months after surgery. Regimens such as FOLFOX, consisting of 5-fluorouracil (5-FU), oxaliplatin, leucovorin, or capecitabine were used for adjuvant chemotherapy. The decision for adjuvant chemotherapy was determined by the TNM stage of the primary cancer, considering the patient’s condition.

### 2.5. Statistical Methods

The chi-square test, Fisher’s exact test, or independent t-test was employed to assess relationships between variables. The normality of the distribution in this study was performed using the Kolmogorov–Smirnov test. Survival analyses for OS and DFS were estimated using the Kaplan–Meier method and the log-rank test. Univariate and multivariate analyses were conducted using the Cox proportional hazards regression model. Time-dependent area-under-the-curve (AUC) and receiver operating characteristic (ROC) curves were assessed. A *p*-value less than 0.05 was considered statistically significant. All statistical analyses were performed using SPSS Statistics 26 (SPSS Inc., Chicago, IL, USA) and SAS version 9.4 (SAS Institute, Cary, NC, USA).

## 3. Results

In this study, we compared the outcomes of patients between Group 0–1 and Group 2, assessed according to preoperative NPS and postoperative NPS scores.

### 3.1. Baseline Patient Characteristics

According to preoperative and postoperative NPS, patients were categorized into Group 0–1 and Group 2, as shown in Table 1. Age, sex, body mass index (BMI), and tumor location did not show significant differences between Group 0–1 and Group 2 in both preoperative and postoperative NPS. American Society of Anesthesiologists (ASA) and serum albumin levels of Group 1 in preoperative NPS showed a higher rate than Group 2, while postoperative NPS showed no significant difference between the groups. The numbers of co-morbidities between group 0–1 and group 2 had no significant difference in preoperative NPS, while showed higher rates of group 2 than group 1 in postoperative NPS (*p* = 0.008). There were no significant differences between Group 0–1 and Group 2 for comorbidities, a history of previous abdominal operation, preoperative CEA level, and NLR. However, total cholesterol, LMR, PLR, and PNI showed significant differences between Group 0–1 and Group 2 in both preoperative and postoperative NPS.

### 3.2. Perioperative Clinical Outcomes

There were no significant differences in operation types and methods between groups in both the preoperative and postoperative NPS systems. Anterior resection was the most common operation in all groups, with laparoscopic procedures being the most common overall. However, the open method was performed in 35.7% of Group 0–1 and 42.1% of Group 2 in the preoperative NPS group and 36.0% of Group 0–1 and 39.6% of Group 2 in the postoperative NPS group. Although the operation time showed no differences between Group 0–1 and Group 2 in preoperative NPS (*p* = 0.627), the operation time of Group 2 was marginally longer than that of Group 1 in postoperative NPS (*p* = 0.073). Although the frequency of emergency operation, stoma formation, and intraoperative transfusion were not significantly different between the two groups, the hospital stay of Group 2 was 13.4 days, which was marginally longer than the 10.7 days of Group 1 in the postoperative NPS group. There was no significant difference in postoperative complications. However, the days to the beginning of adjuvant chemotherapy after surgeries in Group 2 were 31.5 days, which was longer than the 26.6 days of Group 1 in the preoperative NPS. Nevertheless, there was no significant difference in the beginning days of adjuvant chemotherapy in the postoperative NPS group. Although there was no significant difference in the completeness of chemotherapy in both groups, the regimen of chemotherapy showed differences between Group 0–1 and Group 2 in the preoperative NPS (*p* = 0.012), as shown in Table 2.

### 3.3. Pathologic Outcomes According to NPS Groups

In the preoperative NPS group, the tumor size of Group 2 was larger than that of Group 0–1 (5.8 cm vs. 4.4 cm, *p* = 0.002). However, in the pathologic grade, Group 2 of the preoperative NPS group showed a higher rate of signet ring cell carcinoma than Group 0–1 (*p* = 0.020). There was no significant difference between Group 0–1 and Group 2 for the depth of tumor (T stage), nodal status (N stage), TNM stage, numbers of harvested lymph nodes, proximal and distal margins of surgical specimens, lymphovascular invasion, and perineural invasion in both the preoperative and postoperative NPS (Table 3).

### 3.4. Postoperative NPS Transition

NPS transition was defined as the changes in groups between preoperative and postoperative NPS. Upstaging transition was defined as the status where patients had a higher group at postoperative NPS compared to preoperative NPS. The downstaging transition was defined as the status of a lower group at postoperative NPS than preoperative NPS. As shown in Figure 1, the rate of downstaging transition was observed in 19.5% (N = 32) of patients, and upstaging transition was observed in 38.4% (N = 63) of patients. In the comparison among patients with NPS transition, age, sex, ASA, preoperative CEA level, operation method, operation time, the frequency of emergency operations, and intraoperative transfusion showed no significant difference among no changes, upstaging, and downstaging of NPS transition. Additionally, there were no significant differences in hospital stay, postoperative complications, days to beginning adjuvant chemotherapy, tumor size, T stage, N stage, and TNM stage among the groups of NPS transition. Therefore, there were no significant differences in clinical variables for NPS transition in this study (Table 4).

### 3.5. Comparison of Survival According to NPS Group

In OS, there were significant differences for NPS groups preoperatively and postoperatively, as shown in Figure 2a. Group 0–1 in the postoperative NPS showed longer OS compared to group 0–1 in the preoperative NPS group (*p* = 0.022). Both Group 2 of preoperative and postoperative NPS showed poorer overall survival than Group 0–1. On the other hand, DFS between groups in preoperative and postoperative NPS groups did not show significant differences (*p* = 0.731).

### 3.6. Logistic Regression for Survival Outcomes

The prognostic factors affecting OS and DFS were assessed, as shown in Table 5. In OS, the hazard ratio of the incompleteness of adjuvant chemotherapy was 5.321 compared with the completeness of chemotherapy (*p* < 0.001). However, no other clinical factors affected OS. On the other hand, the hazard ratio of DFS in patients with ASA ≥ 2 was 0.312 (*p* = 0.037). However, other clinical factors did not show significant prognostic factors for DFS. Both preoperative and postoperative NPS were not statistically significant for OS and DFS. However, postoperative NPS was considered marginally significant as a prognostic factor for OS (*p* = 0.098).

### 3.7. ROC Curve and AUC for Overall Survival

In the comparison of AUC for preoperative NPS, NLR, PLR, and postoperative NPS, postoperative NPS showed the highest AUC for OS. The AUC of postoperative NPS was 0.64, preoperative NPS 0.57, preoperative NLR 0.52, and preoperative PLR 0.51, as shown in Figure 3a. In the dynamic AUC for 2-year and 5-year OS, the dynamic AUC for 2-year OS of postoperative NPS was 0.65, which was the highest value compared to preoperative NPS, NLR, and PLR. In the 5-year OS, the dynamic AUC of postoperative NPS was 0.64, while preoperative NPS was 0.57 (Table 6).

In the ROC curve for 2-year and 5-year OS, postoperative NPS showed the highest sensitivity compared to preoperative NPS, NLR, and PLR (Figure 3b,c).

## 4. Discussion

Systemic inflammatory and nutritional markers are useful for predicting the prognosis of CRC patients after surgeries. Although many previous studies evaluated the preoperative status, this study demonstrates that postoperative NPS can be useful as a predictive systemic inflammatory marker in stage II–III CRC patients receiving adjuvant chemotherapy compared to preoperative systemic inflammatory markers.

Systemic inflammation plays a crucial role in the prognosis after CRC surgeries. It contributes to the formation of a tumor microenvironment that could participate in tumor cell proliferation, metastasis, angiogenesis, disruption of antitumor immunity, and resistance to anticancer therapy. Cancer cells express various cytokines, chemokines, and growth factors such as IL-6, CCL2, CXCL8, CSF1, and CSF2, which impact vascular permeability, lymph angiogenesis, and metastasis. Moreover, these inflammatory molecules influence cancer cell adhesion and stromal invasion at metastatic sites [19]. On the other hand, it is known that the reversal of systemic inflammation occurs when performing primary resection in CRC patients [20]. Primary resection reduces the inflammatory microenvironment produced by cancer. Furthermore, the occurrence of postoperative complications and recovery after surgeries are strongly related to systemic inflammation and the patient’s prognosis. In this regard, because preoperative inflammatory markers had some flaws in assessing postoperative status to affect prognosis, postoperative NPS holds significance as a predictor with higher accuracy than preoperative NPS.

In this study, we demonstrated the importance of postoperative status through the assessment of NPS transitions after CRC surgeries, as shown in Figure 1 and Table 4. While most studies have focused on preoperative status, it is necessary to consider postoperative complications, such as anastomotic leaks, which can impact postoperative recovery, oral intake, immunosuppressive status, and delayed adjuvant chemotherapy. In this regard, it is known that postoperative complications affect poor oncologic outcomes after CRC surgeries [21]. Arnarson et al. [22] showed that postoperative complications for CRC affect 5-year OS and 3-year DFS. Similarly, in a phase III randomized trial, Aoyama et al. [23] demonstrated that postoperative complications could be an important risk factor affecting OS, DFS, and increasing recurrence rate. Even in the early onset of CRC, it is necessary to assess cancer progression using useful markers [24]. In addition, postoperative infectious complications can affect OS and DFS. According to the study of Artinyan et al. [25], OS and DFS were measured lower in the patients with infectious complications. There were also efforts to find molecular and signal pathways to predict cancer recurrence [26,27] and to develop techniques to prevent postoperative complications [28,29,30] because it is important to maintain curative status and predict recurrence after colorectal surgeries.

The patient’s nutritional status, as well as systemic inflammation, play an important role in predicting the patient’s prognosis after surgeries. Serum albumin level is a factor indicating nutritional status and the degree of inflammation in patients [6,8,31,32]. However, because it tends to change fluidly according to the patient’s liver function or body fluid status, there are limitations to assessing serum albumin alone instead of the inflammation of CRC [9]. Total cholesterol indicates the patient’s nutritional status. The shift of total cholesterol affects the stability and the fluidity of the cell membrane, leading to a decrease in the function of the receptor. Thus, it can lead to a decrease in transmembrane signal transmission function, resulting in difficulty in recovery due to the immune reaction represented by lymphocytes [6,7,33]. Because the nutritional status of patients is changeable after CRC surgeries, the recovery from primary tumor resection, bowel habit changes, and adaptation to the new anastomosis of the bowel should be reflected to predict survival outcomes [16]. In this regard, NPS can be a useful scoring system to predict the prognosis, reflecting both inflammatory and nutritional status simultaneously. Additionally, the higher AUC of postoperative NPS might reflect the higher accuracy of patients’ recovery and their influences on survival outcomes compared with preoperative assessment.

In previous studies, formulations of scoring systems had been suggested for predicting the prognosis, such as CRP, NLR, PLR, LMR, and mGPS (modified Glasgow prognostic score) [34,35]. The CONUT score and PNI, reflecting the nutritional status, are known as useful methods to predict the patient’s prognosis. Although these scoring systems reflect inflammatory biomarkers or nutritional status, they have disadvantages that cannot reflect the changes in the immune system as well as systemic inflammation, which may change after surgeries and the individual patient’s nutritional status. As the enhanced recovery after surgery pathway in CRC patients is based on reflecting the patient’s condition and reducing postoperative complications, it is important to reflect these nutritional statuses to improve the assessment [36,37,38]. Thus, postoperative NPS, reflecting the patient’s postoperative condition, was more useful in predicting the patient’s prognosis than preoperative inflammatory scoring systems in our study.

It is known that minimally invasive surgeries in CRC treatment offer advantages in reducing systemic inflammation and postoperative pain, contributing to fast recovery. In this study, the comparison of NPS among the open, laparoscopic, and robotic surgeries did not show a significant difference between group 0–1 and group 2, both in preoperative and postoperative NPS. Nonetheless, the rapid recovery associated with minimally invasive surgery can improve systemic inflammatory and nutritional status postoperatively. In a meta-analysis comparing laparoscopic and open colorectal surgeries, laparoscopic surgery demonstrated lower levels of CRP and IL-6 compared to open surgery [39]. Additionally, the implementation of ERAS protocol to minimize surgical stress response through minimally invasive surgery has shown an association with improved long-term cancer-specific survival in CRC patients [40]. It is expected that a prospective, large-cohort study utilizing the ERAS protocol in combination with minimally invasive surgeries can enhance the assessment of the NPS system and perioperative assessment to predict oncologic outcomes in CRC patients.

The survival analysis between Group 0–1 and Group 2 showed similar results in both preoperative and postoperative NPS. The OS of Group 0–1 was longer than that of Group 2. The longer OS of Group 0–1 in both preoperative and postoperative NPS suggests the importance of maintaining the patient’s nutritional status and inflammatory status of the primary cancer. However, as the completeness of adjuvant chemotherapy was regarded as the prognostic factor for OS, postoperative management after CRC surgeries is also crucial to improve survival. Because this study included stage II–III CRC patients, cancer status, including obstructive lesions and cancer infiltration, can influence postoperative inflammation and prognosis [41,42]. As shown in the ROC curves, the highest sensitivity of postoperative NPS compared with other scoring systems suggests that the patient’s prognosis is influenced by multidisciplinary aspects of the biology of the primary tumor, the patient’s nutritional status, and adequate postoperative management [43,44,45].

The current study has several limitations to conclude the importance of postoperative NPS in assessing CRC patients. First, it was conducted as a single-center, retrospective study with a limited number of study populations. Selection bias was not controlled using statistical methods such as propensity score matching [46]. Second, tumor location is influenced by different molecular characteristics and prognosis [47,48]. Because we assessed retrospective data, limitations depending on tumor location, genetic mutations, and the severity of tumor progression existed. Third, there was a difference in the timing of the onset of chemotherapy after surgery between Group 0–1 and Group 2. Selection bias and uncontrolled adjuvant treatment can affect the study results. Fourth, the treatment strategies for stage II–III CRC patients have changed. The different characteristics of cancer stages and the old study period of this study can be limitations to validate our results.

However, this study has the strength to elucidate the importance of postoperative inflammatory and nutritional status as prognostic factors after CRC surgeries. Previous studies had limitations in assessing the preoperative tumor status and the patient’s condition prior to surgeries [14,49]. However, because postoperative status can affect oncologic outcomes, our study highlighted the importance of assessing postoperative NPS as a predictor, in contrast to preoperative inflammatory assessment. In future studies, it is necessary to investigate the differences between preoperative and postoperative statuses while controlling study bias in patient selection and postoperative management with a prospective, large cohort, multi-centered study design to achieve comprehensive results and to validate this study’s results.

## 5. Conclusions

NPS is appropriate for predicting the prognosis in stage II–III CRC patients receiving adjuvant chemotherapy following curative resection. Additionally, postoperative NPS showed higher accuracy than preoperative NPS. Predictions based on postoperative clinical outcomes, as well as systemic inflammation of tumors, might be more helpful to assess the prognosis of CRC patients after surgeries.

## Figures and Tables

**Figure 1 cancers-15-05098-f001:**
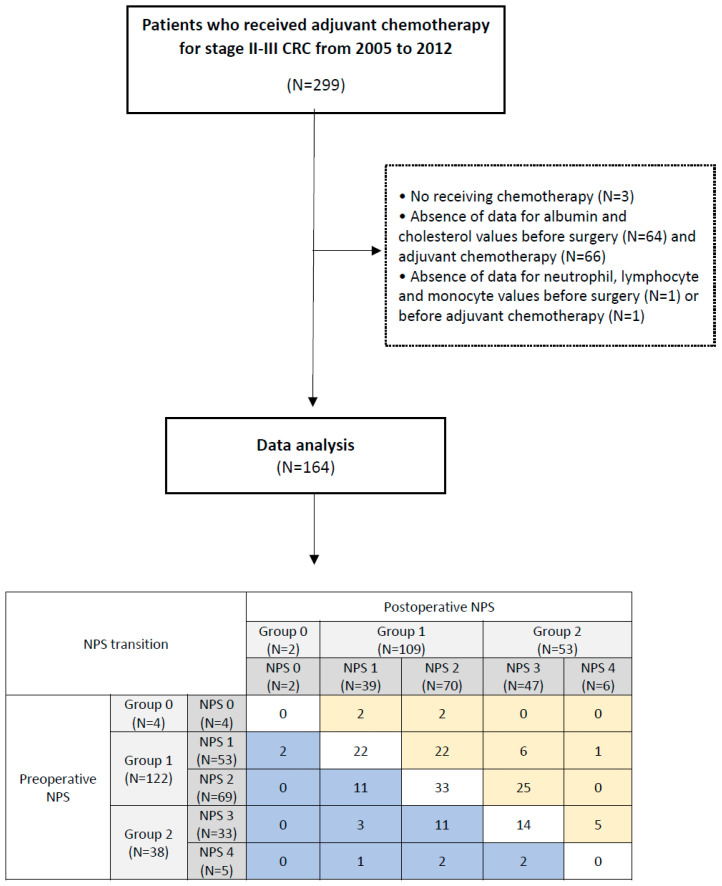
Flow chart of this study.

**Figure 2 cancers-15-05098-f002:**
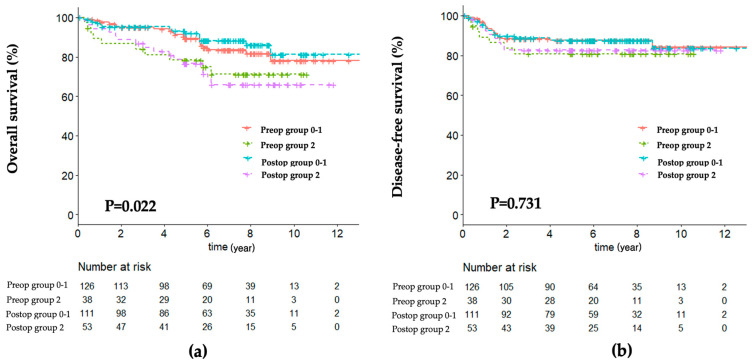
Survival outcomes: (**a**) overall survival; (**b**) disease-free survival.

**Figure 3 cancers-15-05098-f003:**
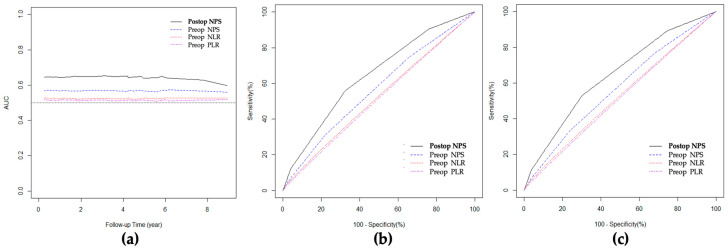
Time-dependent AUC and ROC curves for overall survival: (**a**) time-dependent AUC graph; (**b**) ROC curve (2 year); (**c**) ROC curve (5 year).

**Table 1 cancers-15-05098-t001:** Baseline patient characteristics.

Variables	Preoperative NPS			Postoperative NPS		
	Group 0–1 (N = 126)	Group 2 (N = 38)	*p* Value	Group 0–1 (N = 111)	Group 2 (N = 53)	*p* Value
Age, years ^†^			0.248			0.487
<65	86 (68.3)	22 (57.9)		71 (64.0)	37 (69.8)	
≥65	40 (31.7)	16 (42.1)		40 (36.0)	16 (30.2)	
Sex ^†^			0.189			0.091
Male	69 (54.8)	26 (68.4)		59 (53.2)	36 (67.9)	
Female	57 (45.2)	12 (31.6)		52 (46.8)	17 (32.1)	
BMI	23.2 ± 2.9 (16.5–29.9)	22.9 ± 3.0 (15.3–28.6)	0.633	23.2 ± 3.0 (15.3–29.9)	23.0 ± 2.9 (16.5–28.6)	0.653
Tumor location ^‡^			0.153			0.811
Ascending colon cancer	24 (19.1)	11 (28.9)		26 (23.4)	9 (17.0)	
Transverse colon cancer	3 (2.4)	2 (5.3)		3 (2.7)	2 (3.8)	
Descending colon cancer	9 (7.1)	4 (10.5)		7 (6.3)	6 (11.3)	
Sigmoid colon cancer	50 (39.7)	17 (44.8)		44 (39.6)	23 (43.4)	
Rectosigmoid junction cancer	20 (15.9)	1 (2.6)		15 (13.5)	6 (11.3)	
Rectal cancer	20 (15.9)	3 (7.9)		16 (14.4)	7 (13.2)	
ASA ^‡^			0.043			0.225
1	77 (61.1)	19 (50.0)		70 (63.1)	26 (49.1)	
2	42 (33.3)	12 (31.6)		33 (29.7)	21 (39.6)	
3	7 (5.6)	7 (18.4)		8 (7.2)	6 (11.3)	
Total numbers of co-morbidity ^‡^			0.174			0.008
0	59 (46.8)	15 (39.5)		56 (50.5)	18 (34.0)	
1	46 (36.5)	13 (34.2)		42 (37.8)	17 (32.1)	
2	18 (14.3)	6 (15.8)		10 (9.0)	14 (26.4)	
≥3	3 (2.4)	4 (10.5)		3 (2.7)	4 (7.5)	
Previous abdominal operations ^‡^			0.105			0.911
Yes	55 (43.7)	11 (28.9)		45 (40.5)	21 (39.6)	
No	71 (56.3)	27 (71.1)		66 (59.5)	32 (60.4)	
Preoperative CEA, ng/mL	8.4 ± 15.8 (0.4–138.3)	7.9 ± 12.6 (0.5–72.6)	0.863	7.4 ± 15.3 (0.4–138.3)	10.1 ± 14.5 (0.5–72.6)	0.298
Serum albumin, mg/dL	4.4 ± 0.3 (3.2–5.1)	3.8 ± 0.6 (2.6–4.8)	<0.001	4.3 ± 0.5 (3.1–5.1)	4.1 ± 0.5 (2.6–4.8)	0.056
Total cholesterol, mg/dL	181.4 ± 37.2 (100–295)	141.0 ± 32.4 (74–224)	<0.001	179.9 ± 39.7 (75–275)	155.5 ± 35.1 (74–295)	<0.001
Neutrophil: lymphocyte ratio (NLR)	2.6 ± 1.8 (0.9–14.1)	3.1 ± 2.1 (0.8–11.7)	0.155	2.1 ± 1.4 (0.5–11.3)	2.4 ± 1.0 (0.9–5.7)	0.110
Lymphocyte: monocyte ratio (LMR)	6.0 ± 2.2 (1.6–14.8)	3.9 ± 1.9 (1.1–9.3)	<0.001	6.02 ± 2.15 (2.1–13.7)	4.4 ± 1.6 (1.7–8.9)	<0.001
Platelet: lymphocyte ratio (PLR)	171.6 ± 94.6 (0.3–883.5)	207.3 ± 100.4 (83.5–541.2)	0.046	161.1 ± 75.7 (36.0–470.5)	189.8 ± 77.3 (87.8–373.9)	0.026
Prognostic nutritional index (PNI)	43.9 ± 3.4 (32.0–51.0)	37.7 ± 5.6 (26.0–48.0)	<0.001	42.2 ± 3.6 (32.01–50.01)	39.9 ± 4.3 (30.01–49.01)	<0.001

NPS, Naples prognostic score; BMI, body mass index; ASA, American Society of Anesthesiologists; CEA, carcinoembryonic antigen; N (%); mean ± standard deviation (range); ^†^ Fisher’s exact test; ^‡^ chi-square test.

**Table 2 cancers-15-05098-t002:** Perioperative clinical outcomes.

Variables	Preoperative NPS			Postoperative NPS		
	Group 0–1 (N = 126)	Group 2 (N = 38)	*p* Value	Group 0–1 (N = 111)	Group 2 (N = 53)	*p* Value
Operation names ^‡^			0.168			0.734
Rt. Hemicolectomy	26 (20.6)	12 (31.6)		28 (25.2)	10 (18.9)	
Lt. hemicolectomy	10 (7.9)	7 (18.4)		10 (9.0)	7 (13.2)	
Anterior resection	48 (38.1)	12 (31.6)		41 (36.9)	19 (35.8)	
LAR	38 (20.2)	5 (13.2)		29 (26.1)	14 (26.4)	
Ultra LAR with ISR	1 (0.8)	0 (0)		1 (0.9)	0 (0)	
Segmental resection of colon	1 (0.8)	0 (0)		0 (0)	1 (1.9)	
Hartmann’s operation	1 (0.8)	1 (2.6)		1 (0.9)	1 (1.9)	
Extended resection of colon	1 (0.8)	1 (2.6)		1 (0.9)	1 (1.9)	
Operation method ^‡^			0.774			0.825
Open	45 (35.7)	16 (42.1)		40 (36.0)	21 (39.6)	
Laparoscopy	63 (50.0)	17 (44.7)		56 (50.5)	24 (45.3)	
Robot	18 (14.3)	5 (13.2)		15 (13.5)	8 (15.1)	
Operation time, min	235.3 ± 76.3 (87–473)	228.1 ± 92.9 (97–608)	0.627	225.9 ± 71.5 (87–412)	249.9 ± 94.6 (120–608)	0.073
Emergency operation ^†^	2 (1.6)	1 (2.6)	0.549	2 (1.8)	1 (1.9)	1.000
Intestinal obstruction ^†^	28 (22.2)	14 (36.8)	0.090	31 (27.9)	11(20.8)	0.347
Intestinal perforation ^†^	2 (1.6)	2 (5.3)	0.230	2 (1.8)	2 (3.8)	0.595
Protective stoma formation ^†^	1 (0.8)	2 (5.3)	0.134	2(1.8)	1 (1.9)	1.000
Intraoperative transfusion ^†^	4 (3.2)	0 (0)	0.574	3 (2.7)	1 (1.9)	1.000
Hospital stay, day	10.9 ± 6.8 (5–42)	13.7 ± 11.7 (4–53)	0.070	10.7 ± 6.7 (4–35)	13.4 ± 10.7 (4–53)	0.057
Postoperative complications ^‡^			0.508			0.402
Grade I	9 (7.1)	2 (5.3)		10 (9.0)	1 (1.9)	
Grade II	5 (4.0)	0 (0.0)		3 (2.7)	2 (3.8)	
Grade IIIa	7 (5.6)	2 (5.3)		5 (4.5)	4 (7.5)	
Wound dehiscence						
Grade IIIb	2 (1.6)	2 (5.3)		2 (1.8)	2 (3.8)	
Anastomosis leakage						
Grade IV	0 (0.0)	0 (0.0)		0 (0.0)	0 (0.0)	
Grade V	0 (0.0)	0 (0.0)		0 (0.0)	0 (0.0)	
Days to beginning adjuvant chemotherapy after surgery	26.6 ± 8.0 (10–50)	31.5 ± 11.6 (12–55)	0.003	28.1 ± 8.5 (10–52)	26.9 ± 10.5 (11–55)	0.469
Chemotherapy completeness ^†^			0.331			0.828
Yes	106 (84.1)	29 (76.3)		92 (82.9)	43 (81.1)	
No (Early cessation)	20 (15.9)	9 (23.7)		19 (17.1)	10 (18.9)	
Chemotherapy regimen ^‡^			0.012			0.677
FOLFOX	103 (81.7)	24 (63.2)		87 (78.4)	40 (75.5)	
Capcitabine	23 (18.3)	14 (36.8)		24(21.6)	13 (24.5)	
5-FU/Leucovorin	0 (0.0)	0 (0.0)		0 (0.0)	0 (0.0)	
Others	0 (0.0)	0 (0.0)		0 (0.0)	0 (0.0)	

LAR, low anterior resection; ISR, intersphincteric resection; N (%); mean ± standard deviation (range); NPS, Naples prognostic score; ^†^ Fisher’s exact test; ^‡^ chi-square test.

**Table 3 cancers-15-05098-t003:** Pathologic outcomes according to NPS groups.

Variables	Preoperative NPS			Postoperative NPS		
	Group 0–1 (N = 126)	Group 2 (N = 38)	*p* Value	Group 0–1 (N = 111)	Group 2 (N = 53)	*p* Value
Tumor size, cm	4.4 ± 2.2 (0–14.0)	5.8 ± 3.0 (0.50–13.0)	0.002	4.6 ± 2.5 (0–14.0)	5.0 ± 2.4 (0–13.0)	0.408
Depth of tumor ^‡^			0.757			0.613
pT1	8 (6.3)	2 (5.3)		7 (6.3)	3 (5.7)	
pT2	8 (6.3)	1 (2.6)		7 (6.3)	2 (3.8)	
pT3	83 (65.9)	28 (73.7)		77 (69.4)	34 (64.2)	
pT4	27 (21.4)	7 (18.4)		20 (18.0)	14 (26.4)	
Lymph node metastasis ^‡^			0.168			0.103
pN0	22 (17.5)	12 (31.6)		23 (20.7)	11 (20.8)	
pN1	66 (52.4)	16 (42.1)		61 (55.0)	21 (39.6)	
pN2	38 (30.2)	10 (26.3)		27 (24.3)	21 (39.6)	
TNM stage ^†^			0.112			1.000
Stage II	23 (18.3)	12 (31.6)		24 (7.4)	11 (20.8)	
Stage III	103 (81.7)	26 (68.4)		87 (15.3)	42 (79.2)	
Numbers for harvested lymph nodes	24.8 ± 14.0 (1–71)	28.3 ± 18.2 (0–87)	0.276	26.9 ± 15.6 (0–87)	22.9 ± 13.8 (6–67)	0.118
Proximal resection margin, cm	11.7 ± 7.7 (2–60)	13.9 ± 9.1 (2.5–44)	0.178	12.1 ± 7.9 (2–60)	12.4 ± 8.4 (3–44)	0.801
Distal resection margin, cm	7.2 ± 6.6 (0.5–32)	7.9 ± 5.3 (1–25)	0.495	7.2 ± 6.3 (0.5–32)	7.2 ± 6.5 (0.5–31)	0.759
Pathologic grade ^‡^			0.020			0.674
Well-differentiated	9 (7.1)	3 (7.9)		9 (8.1)	3 (5.7)	
Moderate differentiated	101 (80.2)	27 (71.0)		87 (78.4)	41 (77.3)	
Poor differentiated	5 (4.0)	3 (7.9)		6 (5.4)	2 (3.8)	
Mucinous differentiated	11 (8.7)	2 (5.3)		8 (7.2)	5 (9.4)	
Undifferentiated	0 (0.0)	0 (0.0)		0 (0.0)	0 (0.0)	
Signet ring cell	0 (0.0)	3 (7.9)		1 (0.9)	2 (3.8)	
Lymphovascular invasion ^†^	43 (34.1)	17 (44.7)	0.253	39 (35.1)	21 (39.6)	0.606
Perineural invasion ^†^	21 (16.7)	6 (15.8)	1.000	14 (12.6)	13 (24.5)	0.071

N (%); mean ± standard deviation (range); NPS, Naples prognostic score; ^†^ Fisher’s exact test; ^‡^ chi-square test.

**Table 4 cancers-15-05098-t004:** Details for NPS score transition group.

	No Changes NPS (N = 69)	Upstaging NPS (N = 63)	Downstaging NPS (N = 32)	*p* Value
Age, years ^‡^				0.019
<65	43 (62.3)	49 (77.8)	16 (50.0)	
≥65	26 (37.7)	14 (22.2)	16 (50.0)	
Gender ^‡^				0.974
Male	40 (58.0)	37 (58.7)	18 (56.3)	
Female	29 (42.0)	26 (41.3)	14 (43.8)	
ASA ^‡^				0.990
1	40 (58.0)	37 (58.7)	19 (59.4)	
≥2	29 (42.0)	26 (41.3)	13 (40.6)	
Preoperative CEA level	7.3 ± 11.7 (0.4–72.6)	10.5 ± 20.1 (0.5–138.3)	6.0 ± 8.2 (0.5–42.7)	0.326
Operation method				0.900
Open	27 (39.1)	22 (34.9)	12 (37.5)	
Laparoscopy	34 (49.3)	30 (47.6)	16 (50.0)	
Robot	8 (11.6)	11 (17.5)	4 (12.5)	
Operation time, min	232.1 ± 86.4 (87–608)	240.7 ± 78.0 (130–473)	223.2 ± 70.9 (97–370)	0.595
Emergency operation ^‡^				0.829
Yes	1 (1.4)	1 (1.6)	1 (3.1)	
No	68 (98.6)	62 (98.4)	31 (96.9)	
Intraoperative transfusion ^‡^				0.359
Yes	3 (4.3)	1 (1.6)	0 (0.0)	
No	66 (95.7)	62 (98.4)	32 (100.0)	
Hospital stay, day	11.4 ± 9.3 (4–53)	12.4 ± 7.9 (4–35)	10.4 ± 6.4 (4–31)	0.547
Postoperative complications				0.617
Yes	12 (17.4)	11 (17.5)	8 (25.0)	
No	57 (82.6)	52 (82.5)	24 (75.0)	
Days to beginning adjuvant chemotherapy after surgery	29.8 ± 8.9 (11–55)	24.7 ± 8.4 (10–50)	29.1 ± 9.8 (12–52)	0.249
Tumor size, cm	4.4 ± 2.2 (0.5–10.0)	4.8 ± 2.5 (0.5–14.0)	5.3 ± 2.9 (0.5–12.0)	0.201
TNM stage ^‡^				0.562
Stage II	13 (18.8)	13 (20.6)	9 (28.1)	
Stage III	56 (81.2)	50 (79.4)	23 (71.9)	
T stage ^‡^				0.562
pT1-2	9 (13.0)	5 (8.0)	5 (15.6)	
pT3-4	60 (87.0)	58 (92.0)	27 (84.4)	
Nodal involvement ^‡^				0.465
pN0	12 (17.4)	13 (20.6)	9 (28.1)	
pN1-2	57 (82.6)	50 (79.4)	23 (71.9)	

N (%); Mean ± standard deviation (range); NPS, Naples prognostic score; ^‡^ chi-square test.

**Table 5 cancers-15-05098-t005:** The prognostic factors for overall survival and disease-free survival.

Variables	Event/N (%)	Overall Survival		Event/N (%)	Disease-Free Survival	
		HR(95%CI)	*p* Value		HR(95% CI)	*p* Value
Sex						
Male	22/95 (23.2)	Ref.		14/95 (14.7)	Ref.	
Female	8/69 (11.6)	0.545 (0.237–1.209)	0.133	9/69 (13.0)	0.921 (0.398–2.127)	0.846
Age, yr						
<65	16/108 (14.8)	Ref.		16/108 (14.8)	Ref.	
≥65	14/56 (25.0)	1.844 (0.890–3.820)	0.099	7/56 (12.5)	0.841 (0.346–2.046)	0.702
ASA						
1	16/90 (17.8)	Ref.		16/90 (17.8)	Ref.	
≥2	10/68 (14.7)	0.816 (0.370–1.799)	0.614	4/68 (5.9)	0.312 (0.104–0.934)	0.037
Tumor site						
Right	9/40 (22.5)	Ref.		7/40 (17.5)	Ref.	
Left	16/101 (15.8)	0.679 (0.297–1.553)	0.359	13/101 (12.9)	0.748(0.298–1.874)	0.535
Rectum	5/23 (21.7)	1.055 (0.353–3.153)	0.923	3/23 (13.0)	0.767(0.198–2.967)	0.700
Adjuvant chemotherapy						
Completeness	17/133 (12.8)	Ref.		16/133 (12.0)	Ref.	
Incompleteness	11/28 (39.3)	5.321 (2.451–11.548)	<0.0001	6/28 (21.4)	2.399 (0.938–6.140)	0.068
Preoperative NLR						
<2.96	19/112 (17.0)	Ref.		13/112 (11.6)	Ref.	
≥2.96	11/52 (21.2)	1.173 (0.545–2.524)	0.683	10/52 (19.2)	1.718 (0.753–3.919)	0.198
Preoperative PNI						
<49	28/154 (18.2)	Ref.		23/154 (14.9)	Ref.	
≥49	2/10 (20.0)	1.045 (0.248–4.394)	0.953	0/10 (0.0)	0.294 (0.017–5.147)	0.402
Postoperative NLR						
<2.96	26/140 (18.6)	Ref.		21/140 (15.0)	Ref.	
≥2.96	4/24 (16.7)	0.562 (0.170–1.861)	0.345	2/24 (8.3)	0.491 (0.115–2.099)	0.337
Postoperative PNI			0.601			
<49	30/159 (18.9)	Ref.		23/159(14.5)	Ref.	
≥49	0/5 (0.0)	0.468 (0.027–8.067)		0/5(0.0)	0.624(0.036–10.979)	0.748
Preoperative NPS						
0/1	7/57 (12.3)	Ref.		6/57 (10.5)	Ref.	
≥2	23/107 (21.5)	1.860 (0.757–4.573)	0.176	17/107 (15.9)	1.462 (0.576–3.711)	0.425
Postoperative NPS						
0/1	3/41 (7.3)	Ref.		3/41 (7.3)	Ref.	
≥2	27/123 (22.0)	2.744 (0.830–9.076)	0.098	20/123 (16.3)	2.110 (0.627–7.105)	0.228
NPS transition						
Yes	12/55 (21.8)	Ref.		9/55 (16.4)	Ref.	
No	18/109 (16.5)	0.704 (0.336–1.475)	0.352	14/109 (12.8)	0.814 (0.352–1.881)	0.630
NPS transition						
No change	18/109 (16.5)	Ref.		14/109 (12.8)	Ref.	
Upstaging	3/19 (15.9)	1.020 (0.299–3.483)	0.975	3/19 (15.8)	1.219 (0.350–4.248)	0.756
Downstaging	9/36 (25.0)	1.636 (0.728–3.676)	0.234	6/36 (16.7)	1.234 (0.474–3.212)	0.667

ASA, American Society of Anesthesiologists; LNR, neutrophil–lymphocyte ratio; PNI, prognostic nutritional index; NPS, Naples prognostic score.

**Table 6 cancers-15-05098-t006:** The comparison for AUC and ROC curves.

	Variables	AUC (95% CI)	Pairwise Comparison of *p* Value	
			vs. Preop NPS	vs. Postop NPS
Heagerty’s iAUC	Preop NPS	0.57 (0.50–0.65)	Ref.	
	Postop NPS	0.64 (0.54–0.72)	0.136	Ref.
	Preop NLR	0.52 (0.50–0.62)	0.360	0.032
	Preop PLR	0.51 (0.50–0.62)	0.279	0.027
Heagerty’s incident/ Dynamic AUC (2 year)	Preop NPS	0.57 (0.51–0.65)	Ref.	
	Postop NPS	0.65 (0.55–0.73)	0.114	Ref.
	Preop NLR	0.52 (0.50–0.62)	0.338	0.027
	Preop PLR	0.51 (0.50–0.62)	0.279	0.022
Heagerty’s incident/ Dynamic AUC (5 year)	Preop NPS	0.57 (0.51–0.65)	Ref.	
	Postop NPS	0.64 (0.55–0.73)	0.113	Ref.
	Preop NLR	0.52 (0.50–0.61)	0.339	0.021
	Preop PLR	0.51 (0.50–0.62)	0.261	0.018

## Data Availability

The data presented in this study are available in this article.

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
