# Peer review of "Impact of Postoperative Naples Prognostic Score to Predict Survival in Patients with Stage II–III Colorectal Cancer"

_cancers, 2023, doi:10.3390/cancers15205098_

Round 1

Reviewer 1 Report

It is critical to evaluate the prognosis of patients with colorectal cancer from systemic inflammation and nutritional status. This study aims to evaluate whether postoperative NPS is effective in assessing the prognosis of stage II-III colorectal cancer (CRC) patients. The statistical analysis was well performed and the manuscript was well written.

The following are my review comments:

1.     As the author said, my major concern is the article was a retrospective study with a limited number of samples, and whether the conclusion is extrapolated. It would be better if could verify it with data from other agencies. However, this can be difficult, so the insufficient part of the article needs to be more comprehensive.

2.     Many factors affect the prognosis of patients with CRC, however, the important elements of pathology are not shown in Table 1, such as the degree of differentiation and so on.

3.     I am confused about why also analyzed “Days to beginning adjuvant chemotherapy after surgery” in thePreoperative NPS” in Table 2.

4.     Table 6 refers to the 5-year OS, the AUC of postoperative NPS and preoperative NPS was 0.64 vs. 0.57. Please indicate the P-value for comparison of the 2 data.

5.     In the discussion section, the authors analyzed the significance of this study “…this study has the strength to elucidate the importance of postoperative inflammatory and nutritional status as prognostic factors after CRC surgeries”, however, there must be previous literature analysis on these two points, compared with them, what are the more outstanding highlights of this study? Or what more specific guidance can be provided in clinical practice? It is suggested that the author highlight the meaning of the study in more detail.

6.     There are some small mistakes in the article, which need to be corrected after careful checking. For example, the sentence is incomplete in line 179; “p values” with index p or P, please be consistent.

No.

Author Response

Reviewer #1:

  1. As the author said, my major concern is the article was a retrospective study with a limited number of samples, and whether the conclusion is extrapolated. It would be better if could verify it with data from other agencies. However, this can be difficult, so the insufficient part of the article needs to be more comprehensive.

Response: Thank you for your comments. We agree with your comments regarding the limitations of our study stemming from its retrospective design and the limited number of study participants. We also believe there is a need to analyze and validate our data in a larger-cohort, prospective, multi-center study. In future research, we will strive to conduct a comprehensive analysis and validate the results of this study. We have added sentences addressing this in the discussion on lines 448-449.

  1. Many factors affect the prognosis of patients with CRC, however, the important elements of pathology are not shown in Table 1, such as the degree of differentiation and so on.

Response: Thank you for your comments. In this study, we have presented all pathological elements in Table 3. Based on your comment, including baseline information on pathological characteristics in Table 1 would be helpful to understand the study population. However, given the many pathological factors that influence the prognosis of CRC patients, we believe that presenting all pathological results in Table 3 allows readers to understand the results comprehensively.

  1. I am confused about why also analyzed “Days to beginning adjuvant chemotherapy after surgery” in the“Preoperative NPS” in Table 2.

Response: Thank you for your valuable comments. In this study, we thought that the preoperative NPS might not adequately reflect the postoperative condition of patients, including the timing to initiate adjuvant chemotherapy. Hence, we compared the days taken to commence adjuvant chemotherapy after surgery in both preoperative and postoperative NPS groups. As shown in Table 2, while the preoperative NPS indicated differences between groups 0-1 and group 2 regarding the commencement of adjuvant chemotherapy, the postoperative NPS did not show any significant differences. In the section 2.3, lines 101-105, we have revised the sentences and added explanations regarding the intraoperative outcomes and the timing of initiating adjuvant chemotherapy."

  1. Table 6 refers to the 5-year OS, the AUC of postoperative NPS and preoperative NPS was 0.64 vs. 0.57. Please indicate the P-value for comparison of the 2 data.

Response: Thank you for your comments. We apologize for the confusion regarding the interpretation of Table 6. The P value of the AUC comparing postoperative NPS to preoperative NPS is 0.136. In Table 6, we added 'vs.' to clarify the pairwise comparison for the P value.

  1. In the discussion section, the authors analyzed the significance of this study “…this study has the strength to elucidate the importance of postoperative inflammatory and nutritional status as prognostic factors after CRC surgeries”, however, there must be previous literature analysis on these two points, compared with them, what are the more outstanding highlights of this study? Or what more specific guidance can be provided in clinical practice? It is suggested that the author highlight the meaning of the study in more detail.

Response: The primary strength of this study is its elucidation of the prognostic effects of patients' 'postoperative' status following colorectal surgeries using NPS. Previous studies primarily focused on the 'preoperative' tumor status before surgeries. Galizia et al. evaluated NPS as an independent predictor of long-term outcomes in CRC patients post-surgery (Dis Colon Rectum 2017;60:1273-1284). On the other hand, we compared preoperative NPS, NLR, and PNR with postoperative NPS. As depicted in Figure 1, a downstaging transition occurred in 19.5%, while an upstaging transition was observed in 38.4%. Furthermore, as illustrated in Figure 3, our study highlighted the significance of postoperative NPS in relation to preoperative inflammatory assessments. Consequently, we believe our research underscores the importance of evaluating postoperative status when predicting a patient's prognosis following CRC surgeries. We have incorporated our perspectives into the discussion on lines 441-445.

  1. There are some small mistakes in the article, which need to be corrected after careful checking. For example, the sentence is incomplete in line 179; “p values” with index p or P, please be consistent.

Response: Thank you for your kind comments. We sincerely apologize for the oversights in our sentences. We made corrections to the sentence that was previously on line 179; it is now updated on line 184. To ensure consistency, we've also adjusted the P value notation in our manuscript (line 162). Furthermore, we've standardized the numbering from 'n' to 'N' in Figure 1, Tables 1-4, and the corresponding sentences (lines 227-228). Specifically, in Figure 1, we corrected the number from 'n=6' to 'N=5' for NPS 4 of group 2 in preoperative NPS. We deeply regret these errors. We appreciate you bringing these matters to our attention.

I appreciate the valuable time that was spent in reviewing our paper. Please let us know if there are any additional issues we should address.

Sincerely,

Eun Jung Park MD, PhD, FACS

Reviewer 2 Report

Firstly, I would like to congratulate you by the high quality of the submitted paper. The methodology is excellent, and the information provided has a very high potential clinical relevance.

Its contribution to the international scientific literature could be interesting.

Maybe I would like you to develop more deeply some aspects in your paper. In the following sections, aspects I consider modifiable or revisable of the submitted manuscript will be highlighted.

I continue to present my commentaries to the different parts of the manuscript:

In the INTRODUCTION:

·         Page 2, line 47. Authors must include the original literature cite that described the score NPS they employ.

·         I suggest including in the last sentence that all included patients “that are going to receive standard chemotherapy”.

·         More suggestions on attached PDF document.

Talking about MATERIAL AND METHODS:

·         Figure 1 (associated table), Correction. NPS 3 (n=33)+NPS 4 (n=6) accounts 39 and not 38 as is referred the total N for Group 2: REVIEW AND CORRECT IF NECCESARY.

·         2.4 subsection. Did all or a part of the patients received a preparation or program like ERAS or prehabilitation? Is nutritional state checked systematically in the preoperative period and actions stablished if malnutrition exists?

·         Statistical methods:

o    Which is/are the test/s employed to assess if the variable fulfil normality criteria?

o    Explain for SPSS, as is made for SAS, the factory and its location.

·         More suggestions on attached PDF document.

In the RESULTS SECTION:

·         Line 136: Which comorbidities were assessed? And in the postoperative NPS the p value for comorbidities (assessed a 0, 1 ,2 3 and >3) is p=0.008, that is significant...REVIEW

·         Table 1. Comorbidities expressed as “0, 1, 2 >=3”. Is this the total number? Which ones they consider? Is a comorbidity index? Check that p value is 0.008 and is not reflected on the text.

·         Table 2: It is interesting that minimally invasive surgery, usually associated with less inflammation and a better recovery, seems to not modify the NPS values between preop and postop values, or the changes are similar to the observed in open group. This is an interesting issue for the discussion section, that is not mentioned by authors. CONSIDERE TO ADD TO THE DISCUSSION SECTION.

·         Table 3: Are all of the specimens characterized as R0 (very important)?? In the lymph node metastasis I suggest to add the range (minimum to maximum values) and if there are cases with less than 12 ganglia, reflect the percentage.

·         3.4 subsection, line 222-223. Authors explain PO NPS transitions. What are the suspected reasons to explain this? There were some correcting measures? (as an example: immunonutrition or nutritional support applied preoperatively). CONSIDER TO ADD TO THE DISCUSSION SECTION.

·         3.5 subsection, line 271. Which group? I suppose group 2, NPS 3-4??

·         More suggestions on attached PDF document.

In the DISCUSION SECTION, there are some aspects to be commented deeply:

·         Line 353: postop complications affect oncological outcomes. Some reasons to this event must be presented or hypothesized... Maybe one of the possible reason, for stages that need adjuvant therapies, is because the start in adjuvant therapies may be delayed due to postop complications.

·         See previous commentaries suggesting more issues to add to the discussion in the results section.

·         More suggestions on attached PDF document.

In the CONCLUSIONS SECTION:

·         Line 424: I think there is a mistake. I believe authors must say “following curative resection” (1st surgery and then adjuvant chemo)

Newly I would like to congratulate authors for their work.

Author Response

Reviewer #2:

INTRODUCTION:

  1. Page 2, line 47. Authors must include the original literature cite that described the score NPS they employ.

Response: Thank you for your comments. We added the references for the original literature for NPS.

  1. I suggest including in the last sentence that all included patients “that are going to receive standard chemotherapy”.

Response: Thank you for your kind comments. We corrected the last sentence of introduction as following your comments on line 66.

MATERIAL AND METHODS:

  1. Figure 1 (associated table), Correction. NPS 3 (n=33)+NPS 4 (n=6) accounts 39 and not 38 as is referred the total N for Group 2: REVIEW AND CORRECT IF NECCESARY.

Response: Thank you for your invaluable comments. We've adjusted the count for NPS 4 in group 2 of the preoperative NPS from N=6 to N=5. Furthermore, to maintain consistency throughout the manuscript, we've standardized the numbering from 'n' to 'N'. Figure 1 has also been updated accordingly.

  1. In line 103, please specify TNM edition.

Response: Thank you for your comments. In the revised manuscript, we've included 'American Joint Committee on Cancer (AJCC) the 8th edition' along with the relevant reference.

  1. 4 subsection. Did all or a part of the patients received a preparation or program like ERAS or prehabilitation? Is nutritional state checked systematically in the preoperative period and actions stablished if malnutrition exists?

Response: Thank you for your invaluable comments. Given the retrospective nature of our study, it was not feasible to implement the ERAS or prehabilitation program. In terms of nutritional assessment, the treatment of malnutrition was based on the surgeon's discretion and preferred approach. However, since most CRC patients preferred to have their surgeries as promptly as possible, our primary focus was on performing the surgeries first, followed by attentive postoperative care to aid their recovery. In future prospective studies, we will consider incorporating the ERAS and prehabilitation programs in the treatment of CRC patients.

  1. In line 109, check later if MI surgery proportion is provided and if all resections were R0 and with at least 12 isolated ganglia to obtain a good stratification

Response: Thank you for your comments. We believe that patients who underwent minimally invasive surgeries in this study achieved complete R0 resection. However, the number of harvested lymph nodes varied depending on the surgeons participating in this study. In future research, we will examine the distribution of harvested lymph node numbers in minimally invasive surgeries as you commented.

Statistical methods:

  1. Which is/are the test/s employed to assess if the variable fulfil normality criteria?

Response: We added the sentence detailing the statistical method for normality of distribution on lines 122-123.

  1. Explain for SPSS, as is made for SAS, the factory and its location.

Response: We added the explanation of SPSS for its factory and location on line 128.

RESULTS SECTION:

  1. Line 136: Which comorbidities were assessed? And in the postoperative NPS the p value for comorbidities (assessed a 0, 1 ,2 3 and >3) is p=0.008, that is significant...REVIEW

Response: Thank you for your comments. We categorized patient co-morbidities into several groups: cardiovascular, pulmonary, nephrological, hepatological, diabetic diseases, and other conditions. As some patients had multiple co-morbidities, we counted total numbers of underlying diseases.

  1. Table 1. Comorbidities expressed as “0, 1, 2 >=3”. Is this the total number? Which ones they consider? Is a comorbidity index? Check that p value is 0.008 and is not reflected on the text.

Response: Thank you for your comments. The term of 'co-morbidity' denotes the cumulative numbers of a patient's co-morbidities. When we express co-morbidities as “0,1,2, ≥3”, it indicates the number of co-morbidities a patient has. After categorizing the co-morbidities into six distinct types—cardiovascular, pulmonary, nephrological, hepatological, diabetic diseases, and other conditions—we evaluated the total co-morbidities for each patient. We have updated Table 1 to reflect this by renaming 'co-morbidity' to 'total numbers of co-morbidity'. Moreover, we have added sentences on lines 141-144 to elucidate the significant differences between group 0-1 and group 2 in the postoperative NPS.

  1. Table 2: It is interesting that minimally invasive surgery, usually associated with less inflammation and a better recovery, seems to not modify the NPS values between preop and postop values, or the changes are similar to the observed in open group. This is an interesting issue for the discussion section, that is not mentioned by authors. CONSIDERE TO ADD TO THE DISCUSSION SECTION.

Response: We appreciate your insights regarding the advantages of minimally invasive surgeries in reducing inflammatory responses and enhancing postoperative recovery. Several recent articles have reported the association between the benefits of minimally invasive surgery and improved oncologic outcomes. Although our study did not reveal significant differences among the operative methods, we attribute this to the limitations imposed by the small sample size and the retrospective nature of our study. Thus, we have expanded on these aspects by incorporating a paragraph that covers lines 403-416 within the discussion.

  1. Table 3: Are all of the specimens characterized as R0 (very important)?? In the lymph node metastasis I suggest to add the range (minimum to maximum values) and if there are cases with less than 12 ganglia, reflect the percentage.

Response: Thank you for your comments. We think that all cases in this study achieved R0 resection, as our study focused on evaluating stage II-III CRC patients. As presented in Table 3, we categorized lymph node metastasis as pN0, pN1, and pN2 in accordance with TNM staging criteria. Specifically, pN0 indicates the absence of regional lymph node metastasis, pN1 represents one to three positive regional lymph nodes, and pN2 signifies four or more positive regional lymph nodes. Given that we assessed patient tumor staging based on the TNM stages of CRC, we think that this categorization for positive lymph node can be acceptable to interpret the survival curves according to the NPS in this retrospective study. In future studies, we will incorporate your comment to conduct a more comprehensive evaluation of lymph node metastasis.

  1. 4 subsection, line 222-223. Authors explain PO NPS transitions. What are the suspected reasons to explain this? There were some correcting measures? (as an example: immunonutrition or nutritional support applied preoperatively). CONSIDER TO ADD TO THE DISCUSSION SECTION.

Response: Thank you for your comments. In this study, we tried to show the difference of preoperative and postoperative changes in the patients who received CRC surgeries using NPS transition. Because many previous studies focused on the preoperative status, we deemed it essential to assess the alterations in preoperative and postoperative patient status and their impact on prognosis using the NPS. We have incorporated these reasons into the discussion, as outlined in lines 358-359.

  1. 5 subsection, line 271. Which group? I suppose group 2, NPS 3-4??

Response: It’s group 0-1. We corrected the sentence of line 276-277.

DISCUSSION SECTION:

  1. Line 353: postop complications affect oncological outcomes. Some reasons to this event must be presented or hypothesized... Maybe one of the possible reason, for stages that need adjuvant therapies, is because the start in adjuvant therapies may be delayed due to postop complications.

Response: Thank you for your comments. We agree that postoperative complications, such as anastomotic leaks, postoperative bleeding, and cardiovascular or pulmonary complications, can affect delayed adjuvant chemotherapy and longer hospital stay. The malnutrition and immunosuppressive status resulting from postoperative complications can worsen oncologic outcomes. As you pointed out, we added the sentences in the discussion from lines 360 to 362.

  1. See previous commentaries suggesting more issues to add to the discussion in the results section.

Response: We have carefully reviewed your kind comments and made the necessary corrections. We genuinely appreciate your detailed review and the assistance you provided in completing our manuscript. We have addressed the detailed responses to your comments in the PDF file.

CONCLUSIONS SECTION:

  1. Line 424: I think there is a mistake. I believe authors must say “following curative resection” (1st surgery and then adjuvant chemo)

Response: We appreciate your kind comments. We corrected the sentence: 'NPS is appropriate for predicting the prognosis in stage II-III CRC patients receiving adjuvant chemotherapy following curative resection.'

I appreciate the valuable time that was spent in reviewing our paper. Please let us know if there are any additional issues we should address.

Sincerely,

Eun Jung Park MD, PhD, FACS
